# Inhibitive Mechanism of Loquat Flower Isolate on Tyrosinase Activity and Melanin Synthesis in Mouse Melanoma B16 Cells

**DOI:** 10.3390/biom14080895

**Published:** 2024-07-24

**Authors:** Qianqian Chen, Wenyang Tao, Jianfeng Wang, Jingrui Li, Meiyu Zheng, Yinying Liu, Shengmin Lu, Zhongxiang Fang

**Affiliations:** 1State Key Laboratory for Managing Biotic and Chemical Threats to the Quality and Safety of Agro-Products, Zhejiang Provincial Key Laboratory of Fruit and Vegetables Postharvest and Processing Technology, Ministry of Agriculture and Rural Affairs Key Laboratory of Post-Harvest Handling of Fruits, Institute of Food Science, Zhejiang Academy of Agricultural Sciences, Hangzhou 310021, China; chenqianq2023@163.com (Q.C.); wenyang_tao@163.com (W.T.); ljr990314@163.com (J.L.); 68606459@qq.com (M.Z.); 2College of Life Science, Zhejiang Normal University, Jinhua 321004, China; 3Xingzhi College, Zhejiang Normal University, Lanxi 321100, China; 4School of Agriculture, Food, and Ecosystem Sciences, The University of Melbourne, Parkville, VIC 3010, Australia; lyy19970122@gmail.com (Y.L.); zhongxiang.fang@unimelb.edu.au (Z.F.)

**Keywords:** loquat flower isolate, tyrosinase inhibitory activity, melanin synthesis, melanoma B16 cells, multi-spectroscopy, Western blotting

## Abstract

Melanin naturally exists in organisms and is synthetized by tyrosinase (TYR); however, its over-production may lead to aberrant pigmentation and skin conditions. Loquat (*Eriobotrya japonica* (Thunb.) Lindl.) flowers contain a variety of bioactive compounds, while studies on their suppressive capabilities against melanin synthesis are limited. Loquat flower isolate product (LFP) was obtained by ethanol extraction and resin purification, and its inhibitory efficiency against TYR activity was investigated by enzyme kinetics and multiple spectroscopy analyses. In addition, the impact of LFP on melanin synthesis-related proteins’ expression in mouse melanoma B16 cells was analyzed using Western blotting. HPLC-MS/MS analysis indicated that LFP was composed of 137 compounds, of which 12 compounds, including flavonoids (quercetin, isorhamnoin, *p*-coumaric acid, etc.) and cinnamic acid and its derivatives, as well as benzene and its derivatives, might have TYR inhibitory activities. LFP inhibited TYR activity in a concentration-dependent manner with its IC50 value being 2.8 mg/mL. The inhibition was an anti-competitive one through altering the enzyme’s conformation rather than chelating copper ions at the active center. LFP reduced the expression of TYR, tyrosinase-related protein (TRP) 1, and TRP2 in melanoma B16 cells, hence inhibiting the synthesis of melanin. The research suggested that LFP had the potential to reduce the risks of hyperpigmentation caused by tyrosinase and provided a foundation for the utilization of loquat flower as a natural resource in the development of beauty and aging-related functional products.

## 1. Introduction

Loquat (*Eriobotrya japonica* (Thunb.) Lindl.), belonging to the Eriobotrya of Rosaceae family, is an evergreen fruit tree native to China and is primarily grown in sub-tropical zones [1]. Its flowering season extends from September to the following February and bears fruits in the summertime. Polyphenols including flavonoids are generated in loquat flowers [2]. However, only 1–3% of the flowers can bear fruit, which means that most flowers will wither away after blossoming, resulting in great resource waste. In addition to being used as a traditional Chinese medicine to treat coughs and phlegm, dry loquat flower is now used as a tea-like drink for daily consumption and has been listed in the Chinese new resource food catalogue as approved by the China National Health Commission. It has been reported that loquat flower extracts possess a variety of biological and pharmacological properties including antibacterial, antioxidant, anti-inflammatory, and anti-tumor properties [2,3,4]. However, there is no literature on its tyrosinase inhibitive activity and well-being efficacies on skin such as aberrant pigmentation and skin senescence.

Melanin is a kind of polymer composed of phenolic or indole substances, with complex and heterogeneous chemical structure [5], and it is the main source of complexion, which is produced by melanocytes all over the body. Its synthesis disorder brought by UV radiation, inflammatory reaction, hormones, and senescence, etc., results in pigmentation and other skin conditions [6]. Tyrosinase (TYR), also called polyphenol oxidase, is a multipurpose oxidase contributing to food browning through its oxidative activity and melanin biosynthesis in melanocytes. With a molecular weight of 120 kDa, this metalloenzyme contains two divalent copper ions (CuA and CuB) in its catalytically active pocket [7]. Since it is the only enzyme that limits the rate of melanin production, TYR is crucial to melanin synthesis [7]. Dopaquinone (DQ), directly formed via the oxidation of tyrosine by the action of TYR, undergoes intramolecular cyclization of its amino group to produce cyclodopa (or leucodopachrome), which is then rapidly oxidized by a redox reaction with DQ to give dopachrome and L-3,4-dihydroxyphenylalanine (L-DOPA) [8]. L-DQ is progressively transformed and metabolized into melanin following a sequence of events and rearrangements. Excessive melanin production can result in significant esthetic issues such as hyperpigmentation, dermatosis, and melanoma [2,9]. Therefore, TYR inhibitors have been widely used in pharmaceutical, food, cosmetic, and agricultural products [7,10]. Kojic acid and arbutin are well-known commercial TYR inhibitors that reduce enzymatic reactions causing food browning and skin hyperpigmentation. However, they have numerous side effects, including the risks of cancer, sensitization, and hepatorenal damage if consumed for long time [10]. As a result, new, safe, and effective anti-tyrosinase natural ingredients are urgently needed.

Currently, there is increasing interest in finding natural compounds with melanogenesis inhibitory activity in response to growing demands by people who care for their skin well-being and senescence. Active compounds derived from plants, such as arbutin, aloesin, gentisic acid, flavonoids, licorice, niacinamide, yeast derivatives, and polyphenols, have been found to inhibit melanin production without harming melanocytes through various mechanisms [11]. The melanin inhibitory effects from plants and phytochemicals have been reviewed by Feng, Fang, and Zhang [12] who find flavonoids, phenolic acids, stilbenes, and terpenes are main classes of phytochemicals responsible for the melanin inhibition. The plant extracts/phytochemicals show a melanin inhibitory effect through three mechanisms: the inhibition of tyrosinase, the down-regulation of microphthalmia-associated transcription factor gene expression, and the absorption of UV radiation. The methanolic extracts of *Dimorphandra gardneriana*, *Dimorphandra gardneriana*, *Lippia microphylla*, and Schinus terebinthifolius are the most potent ones worth further investigation in animal and/or human trials [12].

As far as we know, there is no report on the inhibition activity of loquat flower extract on tyrosinase activity and melanin synthesis at present. In this study, the tyrosinase inhibitory constituents in loquat flowers were extracted and resin-isolated to obtain loquat flower isolate product (LFP), and its influence on in vitro TYR activity and action mode were investigated using UV-Vis, FT-IR, and fluorescence spectrometry. Mouse melanoma B16 cells were adopted to further investigate the melanin synthesis and its pathway affected by LFP through Western blotting. The results could provide a scientific foundation for the development of loquat flower extract or isolate as a whitening agent or ingredient in functional foods and cosmetics.

## 2. Materials and Methods

### 2.1. Chemicals and Materials

Tyrosinase (EC1.14.18.1, 500 U/mg), L-3,4-Dihydroxyphenylalanine (L-DOPA), kojic acid, sodium hydroxide, copper sulphate, and phosphoric acid buffer (pH 6.8) were obtained from Yuanye Biological Technology Co., Ltd. (Shanghai, China). Methanol and acetonitrile of HPLC-MS grade were bought from Merck KgaA (Darmstadt, Germany), and formic acid (HPLC-MS grade) was from Xiya Chemical Technology Co., Ltd. (Linyi, Shandong, China). D101 macroporous resin was supplied from Macklin Biochemical Technology Co., Ltd. (Shanghai, China). Mouse melanoma cell line B16 was purchased from Haotian Biotechnology Co., Ltd. (Hangzhou, China). DMEM complete culture medium, fetal bovine serum (FBS), penicillin-streptomycin solution (100×), proteinase inhibitor, trypsase, and Cell Counting Kit-8 (CCK8) reagent were purchased from Beijing Labgic Technology Co., Ltd. (Beijing, China). Tyrosinase-related protein (TRP) 1 and 2, sheep anti-rabbit second antibody, and sheep anti-mouse second antibody were obtained from Affinity Biosciences (Cincinnati, OH, USA). Deionized water was used through the experiment. Loquat dry flowers (cv. Ninghai Bai, 5% in moisture content) were supplied by Meiqi Biotechnology Co., Ltd. (Ninghai, Zhejiang, China). The samples were obtained by microwave drying at 700 W for 6–8 min and identified by Dr./Prof. Chen Junwei, a pomologist engaged in loquat variety breeding and cultivation in the Zhejiang Academy of Agricultural Sciences.

### 2.2. Preparation of LFP Sample

Using TYR inhibitory activity as the index, loquat dry flowers were extracted twice with 50% ethanol (1:20 g/mL in material-liquid ratio, 2 h, and 50 °C) after being ground and sieved through a 50-mesh screen. After being pooled, extract solutions were centrifuged by a centrifuge (LXJ-IIB, Anting Scientific Instrument Factory, Shanghai, China) at 4000× *g* at room temperature, and the supernatant was collected. The crude extract was purified using D101 macroporous resin at a loading concentration of 40 mg/mL, a loading flow rate of 3.0 BV/h, an eluent concentration of 60% ethanol, and an elution flow rate of 4.0 BV/h. The eluate was concentrated in a rotary evaporator (RE-2000A, Yarong Co., Ltd., Shanghai, China) and then lyophilized using a freeze drier (FD-1A-50, Xinzhi Co., Ltd., Ningbo, China) to obtain LFP.

### 2.3. Identification of Constituents in LFP

The LFP powder was dissolved in 70% methanol to make a solution of 100 mg/mL before high-performance liquid chromatography (HPLC)-mass spectrometry (MS)/MS detection was carried out using a Q-Exactive HF high-resolution mass spectrometer (Thermo Fisher Scientific (China) Co., Ltd., Shanghai, China) equipped with a Zorbax Eclipse C18 chromatographic column (1.8 μm × 2.1 mm × 100 mm, Agilent Technology (China) Co., Ltd., Beijing, China). The HPLC separation conditions were using column temperature at 30 °C and a flow velocity of 0.3 mL/min with mobile phases A and B being 0.1% formic acid aqueous solution and pure acetonitrile, respectively. The injection volume was 2.0 μL, and the automatic sampling temperature was 4 °C. The MS conditions were 3.5 KV in ion spray voltage and 325 °C for ionization temperature. The scan range of the mass was 100–1500 *m*/*z*.

Compound Discoverer 3.3 (Thermo Fisher Scientific Inc., Waltham, MA, USA) was used to perform compounds’ retention time correction, peak identification, and extraction. Based on MS/MS information, the Thermo mzCloud online database and Thermo mzValut local database were adopted to carry out substance identification.

### 2.4. Tyrosinase Inhibitory Activity and Kinetic Type Assays

TYR activity was determined with a MAPAD UV-3100PC spectrophotometer (Meipuda Instrument Co., Ltd., Shanghai, China) linked to an LRH-250 incubator (Shanzhi Co., Ltd., Shanghai, China). In a 96-well plate, 100 μL PBS buffer (pH 6.8), 100 μL substrate (L-DOPA, 200 g/mL), 100 μL TYR (100 U/mL), and 2 μL LFP at various concentrations (0, 1.25, 2.5, 5, 10, 20, and 40 mg/mL) were added and mixed and then incubated at 37 °C for 10 min before the absorbances of the reaction solutions were measured at 475 nm. Kojic acid was used as the positive control.

With the L-DOPA concentration (1 mg/mL) remaining constant in the reaction system, different concentrations of TYR solution (50, 100, 125, and 150 U/mL) and LFP (0, 1.5, 2.5, and 3.5 mg/mL) were used to investigate the type of TYR activity inhibition by LFP. The horizontal *X*-axis was set to the TYR concentration ([E]), and the *Y*-axis was set to the reaction velocity (v). The plotting was used to determine whether the inhibition of TYR by LFP was reversible or not [13].

To further investigate the inhibitive mode, different concentrations of L-DOPA (0.05, 0.075, 0.1, 0.125, 0.15, 0.175, and 0.2 mg/mL) and LFP concentrations (0, 1, 2, and 3 mg/mL) were used in the reaction system. The reciprocals of the substrate concentration (1/[S]) and the reaction rate (1/[v]) were set as the abscissa and ordinate of the Lineweaver–Burk plot, respectively.

### 2.5. Determination of Copper Ion Chelating Ability of LFP

The copper ion chelating ability of LFP was assessed using the method reported by Liu et al. [14]. Briefly, a reaction system was composed of 1.8 mL PBS (pH 6.8), 0.1 mL LFP (2.8 mg/mL), and 0.1 mL CuSO_4_ at various concentrations (0, 5, 10, 15, 20, or 25 mmol/mL). Each group of reaction solution was kept reacting at 37 °C for 10 min before being scanned between 240 and 400 nm on a UV-Vis spectroscopy (UV-3600plus, Shimadzu (China) Co. Ltd., Shanghai, China).

### 2.6. Analysis of the Secondary Structure of Tyrosinase

The FT-IR spectrum was analyzed using the method of Ju et al. [15], with minor modifications. The FT-IR spectra of TYR (100 U/mL in PBS at pH 6.8, incubated at 37 °C for 10 min) in the absence and presence of LFP (2.0 mg/mL) were recorded using a Thermo Nicolet-5700 spectrometer (Thermo Nicolet Corporation, Madison, WI, USA) in the wavenumber range of 4000–400 cm^−1^. The resolution was 5 cm^−1^, and the number of scans was 64.

### 2.7. Analysis of the Conformation Change in Tyrosinase

The reaction mixture (4 mL) included 3 mL PBS (pH 6.8), 900 μL TYR (100 U/mL), and 100 μL LFP with different concentrations (0, 0.5, 1.0, 1.5, 2.0, 2.5, 3.0, 3.5, and 4.0 mg/mL). Each reaction system’s fluorescence absorbance was determined using a spectrofluorometer (model F-7000, Hitachi, Japan) with the excitation wavelength of 275 nm, a 5 nm slit width of the excitation and emission spectra, 600 V of voltage, and 1200 nm/s of scanning speed. Each sample group’s emission spectrum was scanned from 300 nm to 480 nm.

The Stern–Volmer equation for fluorescence quenching is shown below.
(1)$$\frac{F_{0}}{F}=1+K_{q}\tau_{0}\left[Q\right]=1+K_{sv}[Q]$$
where *F*_0_ and *F* represent the fluorescence intensity of TYR in the absence and presence of quencher (LFP), respectively. *K_q_* is the quenching rate constant, and *τ*_0_ is the average lifetime of the fluorophore without quencher. [*Q*] means the concentration of LFP, and *K_sv_* denotes the Stern–Volmer quenching constant.

### 2.8. Cell Viability Assay

The cell viability assay was conducted according to that reported by Xiang et al. [16] with slight modifications. The mouse melanoma B16 cells were cultured in DMEM medium containing 10% FBS and 1% TritonX-100 in a 5% CO_2_ incubator (Forma^TM^ serie II, Thermo Fisher Scientific Inc., Waltham, MA, USA) at 37 °C. In 96-well plates, cells at 1 × 10^4^ cells/mL were inoculated and incubated for 24 h. The medium was then replaced with different concentrations of LFP (12.5, 25, 50, 100, 200, and 400 µg/mL) and incubated for 48 h. After the supernatant discarded, 10 μL of CCK8 solution and 90 μL of serum-free medium were added and incubated for 2 h. Using a microplate reader (Tecan Corporation, Männedorf, Switzerland), the absorbance of the mixture in each well was measured at 490 nm. The viability results were expressed as the percentage of absorbance in sample cells relative to control cells (without LFP). Every sample was measured in triplicate, and each experiment was carried out at least three times.

### 2.9. Determination of Tyrosinase Activity and Melanin Content in B16 Cells

TYR activity and melanin content in the cells were determined using spectrophotometry, referred to those reported by Kim et al. [17] with some modifications. In brief, mouse B16 cells at a density of 6 × 10^5^ cells/mL were seeded onto 6-well plates, and the supernatants were discarded with a pipette after the cells were incubated for 24 h (37 °C, 5% CO_2_). LFP solutions (2 mL) of various concentrations (0, 25, 50, and 100 µg/mL) were then respectively added to the cell precipitates, and the mixtures were incubated for 48 h. The cells were lysed with 1 mL PBS (pH 6.8) containing 1% TritonX-100 using an ultrasonic crusher (JY92-IIN, Scientz Biotechnology Co., Ltd., Ningbo, China), transferred to 96-well plates, and treated with L-DOPA solution (1 mol/mL, 100 μL, 37 °C) for 10 min. The absorbance of the reaction solution at 475 nm was then measured, and the TYR activity was calculated as a percentage of the absorbance in sample treated cells compared with the control (without LFP). Each sample was measured in triplicate, and each experiment was repeated at least three times.

The cultured cells in the sample treated groups and the control were rinsed three times with PBS (pH 6.8), centrifuged at 1000× *g* and 25 °C to remove supernatants before being dissolved in 100 μL of 1 mol/L NaOH containing 10% DMSO, and heated in an 80 °C water bath for 1 h to fully release melanin. The cell lysate was diluted with ultrapure water until it reached a volume of 400 μL. The melanin content in cell lysate was determined at the absorbance of 405 nm and expressed as a percentage of absorbance of the sample group to that of the control. Every sample was measured in triplicate, and each experiment was carried out at least three times.

### 2.10. Western Blotting Analysis

The cultivated cells treated with LFP (0, 50, 100 µg/mL) were used to carry out Western blotting tests. After being washed with PBS (pH = 6.8) and collected by centrifugation as described above, the cells were lysed in a lysis buffer (150 mM NaCl, 10 mM Tris (pH 7.5), 5 mM EDTA, and 1% Triton X-100) containing a protease inhibitor at 4℃ for 20 min. Proteins were resolved by SDS polyacrylamide gel electrophoresis (SDS-PAGE) and electrophoretically transferred to a polyvinylidene fluoride (PVDF) membrane (Merk Millipore, Billerica, MA, USA). The membrane was blocked in 5% fat-free milk in PBST buffer (PBS with 0.05% Tween-20) for 1 h. After a brief wash, the membrane was incubated overnight at 4 °C with several antibodies: anti-TYR (1:1000, *v*/*v*), anti-TRP1 (1:5000), and anti-TRP2 (1:1000). A subsequent incubation with goat anti-mouse antibody (1:7500) conjugated with horseradish peroxidase was conducted at room temperature for 2 h. Specific protein bands were visualized by chemiluminescence using ECL solution (Cytiva, Tokyo, Japan) and detected by ImageQuan LAS 500 (GE Healthcare, Chicago, IL, USA) for the quantification of the band shadow area with Image J software (National Institutes of Health, Bethesda, MD, USA).

### 2.11. Statistical Analysis

All data were represented as mean ± SD. The statistical analysis of the results was performed using one-way ANOVA with Tukey’s correction for multiple comparisons. All data were analyzed using GraphPad Prism9.0 (GraphPad Software Inc., San Diego, CA, USA). 

## 3. Results and Discussion

### 3.1. Constituents of LFP

HPLC-MS/MS analysis indicated that there were 26 classes of compounds and 137 compounds in total in the LFP (Appendix A). Among them, 12 compounds including quercetin, isorhamnoin, *p*-coumaric acid, etc., in the categories of flavonoids and cinnamic acid and its derivatives, as well as benzene and its derivatives (Table 1), were reported to have potential tyrosinase inhibitory activities after a comparison with the literature.

### 3.2. Inhibition Type of LFP on Tyrosinase Activity

The IC50 value, representing the concentration of LFP required to reduce TYR activity by 50%, was used to reflect LFP’s inhibitory ability on TYR. As shown in Figure 1A, the TYR activity decreased as the concentration of LFP increased, with its IC50 being 2.8 mg/mL, while kojic acid (Figure 1B) had an IC50 value of 1.8 mg/mL. Although LFP had a slightly lower inhibitory ability than that of kojic acid, it had the advantages of easy production, low cost, greenness, and safety compared with the latter.

The relationship between the oxidation reaction rate (v) of the substrate and the concentration of TYR [E] affected by LFP is shown in Figure 1C, which determines whether the inhibition of the LFP sample on TYR activity is reversible or not. The result showed that the slope of the straight line was decreased as the LFP concentration increased, and all the lines passed through the origin of the coordinate axes, indicating that the inhibitory effect of LFP on TYR activity was reversible [27].

As the LFP concentration increased, the straight lines in Figure 1D became parallel to each other, indicating that the inhibitory mode of LFP on TYR activity was an anti-competitive one [28]. In contrast to competitive and non-competitive inhibitions, the inhibitor with the anti-competitive one does not directly bind to the enzyme but rather combines with the enzyme–substrate complex to form an enzyme–substrate–inhibitor complex (the complex cannot produce oxidized products), thereby affecting the progress of enzyme catalysis [29]. Due to there being 26 classes of compounds in LFP as shown in Appendix A, the inhibition of LFP to TYR seems complicated and might be the combined effects of all the compounds.

### 3.3. Copper-Ion Chelating Ability of LFP

Increasing evidence suggests that TYR inhibition is dependent in part on the inhibitor’s ability to chelate the copper ions in the active site of the enzyme [30]. The chelating property of LFP to Cu^2+^ was thus investigated. LFP had UV absorption peaks at 242 nm and 246 nm, as shown in Figure 2A. LFP’s absorption peak intensities were decreased after Cu^2+^ was added. However, the characteristic absorption peak positions of LFP did not shift as the concentration of Cu^2+^ increased, and no additional peak appeared, indicating that there was no direct interaction between LFP and Cu^2+^. Although the interactions between Cu^2+^ and some dietary flavonoids including quercetin have been reported to occur in a pH titration solution when UV–Vis spectroscopy was combined with chemometric analysis as applied by Zhang et al. [31], flavonoids in LFP might not directly act on Cu^2+^ in active pockets of TYR. The results suggested that LFP may not inhibit tyrosinase activity through chelating copper ions at the active center of the enzyme.

### 3.4. Effect of LFP on Secondary Structure of Tyrosinase

Fourier transform infrared spectroscopy was used to investigate the effect of LFP on the secondary structure of TYR protein, and the results are shown in Figure 2B. The absorption peaks at 3503 and 3474 cm^−1^ in the protein were attributed to the stretching vibrations of N–H, and the contraction vibrations of -CH_2_ in the enzyme might be responsible for the two absorption peaks at 2190 and 2160 cm^−1^. Bands (1700–1600 cm^−1^) at Amide I of TYR occurred due to the stretching vibrations of C=O on the skeleton of the protein, while those (1600–1500 cm^−1^) at Amide II of the protein were caused by stretching vibrations of C–H and bending one N–H in the peptide chains [32]. 

After being added to the LFP sample solution, the absorption peaks at 2931 and 2855 cm^−1^ in the enzyme might be brought about by saturated C–H stretching vibrations, while those at 2190 and 2160 cm^−1^ were shifted to 2225 and 2167 cm^−1^, respectively, possibly due to unsaturated bonds or aromatic conjugation formed between the enzyme and components in LFP [29]. It was worth noting that the addition of LFP made all the characteristic adsorption peaks in the Amide I and Amide II bands of TYR appear in redshift, which indicated that interaction occurred between the LFP and the enzyme and caused the rearrangement of the hydrogen bonding network on polypeptide chains, resulting in a configuration change in tyrosinase [14].

### 3.5. Effect of LFP on Conformation of Tyrosinase

Tryptophan (Try) and phenylalanine (Phe) are two fluorescent chromophore groups in the TYR protein. Try’s fluorescence intensity is higher than Phe’s because it has one more conjugated double bond. Moreover, Try’s fluorescence spectrum may even cover Phe’s because the two fluorescence spectra overlap [33]. When TYR is analyzed under fluorescence spectrophotometer, there is a fluorescence peak of its own, which will rise or fall depending on whether the inhibitor has interactions with it. Therefore, this phenomenon can be used to assess the effect of inhibitors on tyrosinase conformation. The position of the fluorescence peak in TYR shifted from 332 to 338 nm when LFP was added, as shown in Figure 3A. The redshift in the fluorescence peak of TYR with the increased addition of LFP indicated a decrease in the energy required for TYR to maintain its own structure and a change in its conformation [29]. In addition, the fluorescence intensity of TYR was gradually decreased as the LFP concentration increased (Figure 3B). The results suggested that compounds in LFP, possibly flavonoids, may combine with TYR to quench its intrinsic fluorescence and affect the microenvironment hydrophobicity of fluorescent residues in tyrosinase [34]. The combination could be due to the hydrophobic interaction between flavonoids in LFP and amino acid residues in tyrosinase, resulting in the exposure of protein polypeptide chains to the environment, thereby inhibiting the catalytic reaction of tyrosinase [35]. There was a linear relationship between *F*_0_/*F* and [*Q*] in Figure 3C, and the *K_sv_* value decreased with increasing temperature (296 K to 303 K). As a result, the fluorescence quenching of tyrosinase by LFP was determined to be static quenching [36]. The data suggested that LFP could bind to tyrosinase when tyrosinase and LFP coexisted, and its activity could be significantly inhibited since its fluorescence intensity decreased dramatically as the LFP content increased.

### 3.6. Effect of LFP on the Viability of Mouse Melanoma B16 Cells

The effect of LFP on the viability of mouse melanoma B16 cells was assessed by the CCK8 method [16]. Result showed that LFP solution at less than 200 μg/mL was non-toxic to the cells, with a survival rate of more than 80% (Figure 4A). However, the viability of the cells was decreased in a dose-dependent manner as the concentration of LFP increased. Since an LFP of less than 200 μg/mL had no cytotoxic effect on B16 melanoma cells, 0–200 μg/mL LFP were used in the following experiments.

### 3.7. Effect of LFP on Tyrosinase Activity and Melanin Synthesis in B16 Cells

To evaluate the effects of LFP on melanin production, we measured the intracellular TYR activity and melanin content in B16 cells. At different concentrations (25, 50, and 100 μg/mL), LFP inhibited TYR activity and melanin formation in a positive dose-dependent manner, as shown in Figure 4B. The TYR activity in the cells was significantly decreased (*p* < 0.05) after being treated with 50 μg/mL LFP, and it was further reduced (*p* < 0.01) to 51.6% when the treatment concentration was 100 μg/mL, compared with that of the blank control. This demonstrated that LFP treatment at 50 μg/mL or higher concentration significantly inhibited tyrosinase activity in mouse melanoma B16 cells.

Figure 4C showed LFP treatment (25–100 μg/mL) significantly reduced the melanin content in B16 cells compared with the blank control (*p* < 0.01). The relative contents of intracellular melanin decreased to 82.8% and 56.2% after the cells were treated with 50 and 100 μg/mL LFP, respectively. This indicated that LFP had a significant inhibition effect against melanin synthesis in mouse melanoma B16 cells. These results suggested that LFP could reduce the synthesis of melanin through inhibiting B16 intracellular TYR activity.

### 3.8. Western Blotting Results

As a rate limiting enzyme in melanin synthesis, TYR directly affects the reaction rate of melanin synthesis. TRP2 is involved in the chemical reaction that converts dopamine pigment to 5,6-dihydroxyindole-3-carboxylic acid (DHICA) during melanin synthesis, while TRP1 participates in the transformation of DHICA to 5,6-indolequinone carboxylic acid, as well as the oxidation reaction that produces melanin [37,38,39]. These three proteins are located downstream in the protein pathway of melanin synthesis and are closely related to melanin synthesis. As shown in Figure 5A, there was no significant difference in the expression of TYR after 100 μg/mL LFP treatment compared with 25 μg/mL kojic acid treatment (*p* > 0.05). However, LFP inhibited TYR expression in a positive concentration-dependent manner, and the expression was significantly lowered by 100 μg/mL LFP compared with 50 μg/mL LFP (*p* < 0.01), which was consistent with its TYR activity inhibition trend in Figure 4B. When compared with the blank control, the experimental group treated with 50 μg/mL LFP had significantly reduced expression of TYR (*p* < 0.01), while that treated with 100 μg/mL LFP had an extremely significant decrease in TYR expression (*p <* 0.0001). The findings indicated that LFP at a concentration of more than 50 μg/mL had a significant inhibitory effect on TYR expression in B16 cells.

In addition, TRP1 expression levels were decreased extremely significantly (*p* < 0.0001) and significantly (*p* < 0.01) after 50 and 100 μg/mL LFP were administered to the cells, respectively, as shown in Figure 5B. Particularly, the 50 μg/mL LFP treatment resulted in lower TRP1 expression than the positive control (25 μg/mL kojic acid). TRP1 expression in the 50 μg/mL LFP-treated group was significantly lower than that of the 100 μg/mL LFP-treated group (*p* < 0.01), which needed more investigation to reveal the potential explanation. Furthermore, the TRP2 expression levels were suppressed by LFP treatment in a positive concentration-dependent manner as well (Figure 5C). The TRP2 expression was significantly reduced by both the 50 μg/mL and 100 μg/mL LFP treatments compared with the blank control (*p* < 0.05 and *p* < 0.01, respectively).

These results indicated that LFP inhibited intracellular tyrosinase activity and melanin synthesis primarily through suppressing the expressions of TYR, TRP1, and TRP2. However, due to there being 12 components in LFP with reported TYR inhibitory activities (Table 1), the main contributors in LFP for the inhibitory abilities to TYR activity and melanin synthesis and their potential synergistic/counteractive actions await further in-depth investigation in the future.

## 4. Conclusions

Loquat flower isolate had considerable TYR inhibitory ability with its IC_50_ value of 2.8 mg/mL, which was slightly higher than that of a commercial medicine kojic acid (1.8 mg/mL). LFP inhibited in vitro TYR activity in an anti-competitive inhibition mode but without a chelating effect on copper ions. The inhibition of LFP on TYR activity was achieved by affecting the secondary structure and conformation of the enzyme protein. At the cellular level, it was demonstrated that tyrosinase activity in mouse melanoma B16 cells was significantly inhibited after the cells were treated with LFP at concentrations of 50 μg/mL and 100 μg/mL (without cytotoxicity) (*p* < 0.05 and 0.01). The cells had significantly reduced synthesis rates of melanin after being treated with LFP at 25–100 μg/mL (*p* < 0.01). The Western blotting results indicated that LFP inhibited melanin synthesis through decreasing the expressions of TYR, TRP1, and TRP2. These findings provided a preliminary scientific foundation for the development of loquat flower isolate as a natural tyrosinase inhibitor and potential whitening or antioxidative ingredient in functional foods and cosmetics. More detailed studies about the pathways in melanin synthesis affected by individual main components in LFP and their possible synergistic action are needed.

## Figures and Tables

**Figure 1 biomolecules-14-00895-f001:**
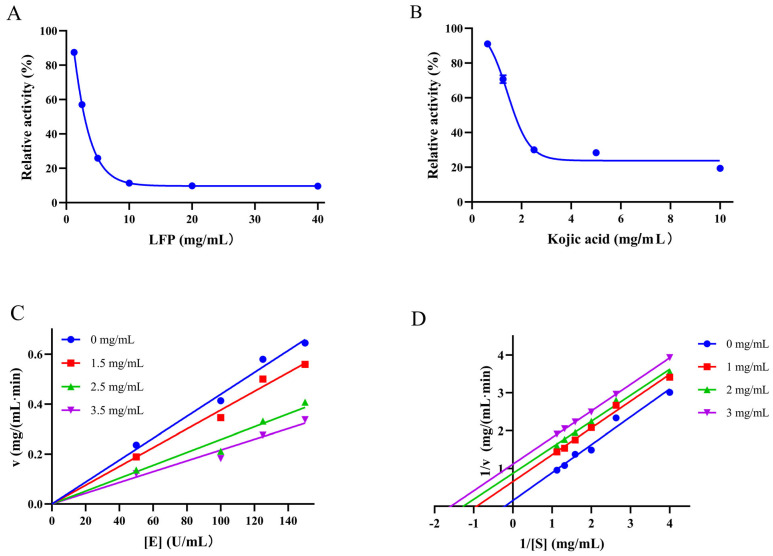
Inhibitory effects of LFP (**A**) and kojic acid (**B**) on in vitro activities of tyrosinase and LFP’s effect on the reaction rate of tyrosinase (**C**) and its Lineweaver–Burk curves (**D**). v on the ordinate and [E] and [S] on the abscissa refer to the reaction rate, tyrosinase concentration, and substrate (L-DOPA) concentration, respectively.

**Figure 2 biomolecules-14-00895-f002:**
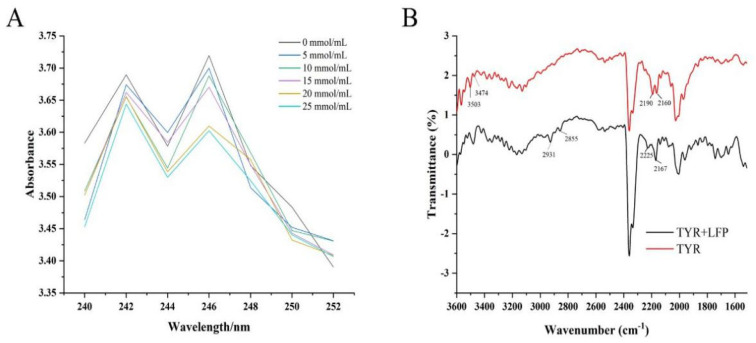
Effects of different concentrations of Cu^2+^ on the ultraviolet-visible spectra of LFP (**A**) and the comparison on the Fourier transform infrared spectra of tyrosinase and its mixture with LFP (**B**).

**Figure 3 biomolecules-14-00895-f003:**
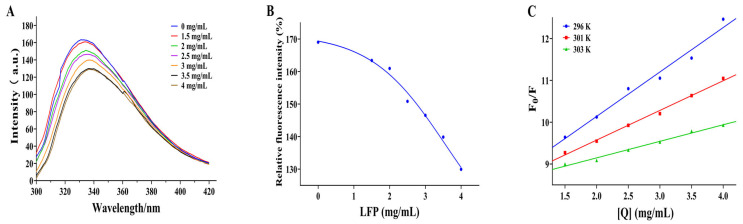
Effect of different concentrations of LFP on the fluorescence intensity of tyrosinase (**A**), the relative fluorescence intensity changed with LFP concentrations (**B**), and Stern–Volmer curve graphs of fluorescence quenching of tyrosinase by LFP (**C**). *F*_0_ and *F* represent the fluorescence intensity of TYR in the absence and presence of LFP, respectively, and [*Q*] means the concentration of the quencher (LFP).

**Figure 4 biomolecules-14-00895-f004:**
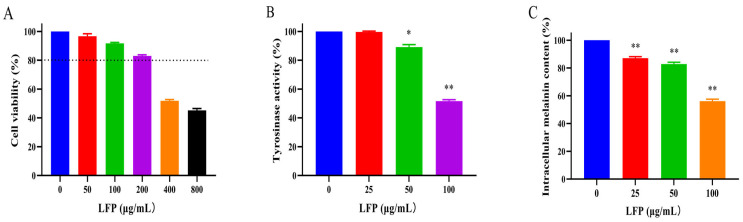
Effect of LFP on the cell viability (**A**), TYR activities (**B**), and melanin contents (**C**) in mouse melanoma B16 cells. * and ** indicate *p* < 0.05 and *p* < 0.01, respectively, compared with the blank control.

**Figure 5 biomolecules-14-00895-f005:**
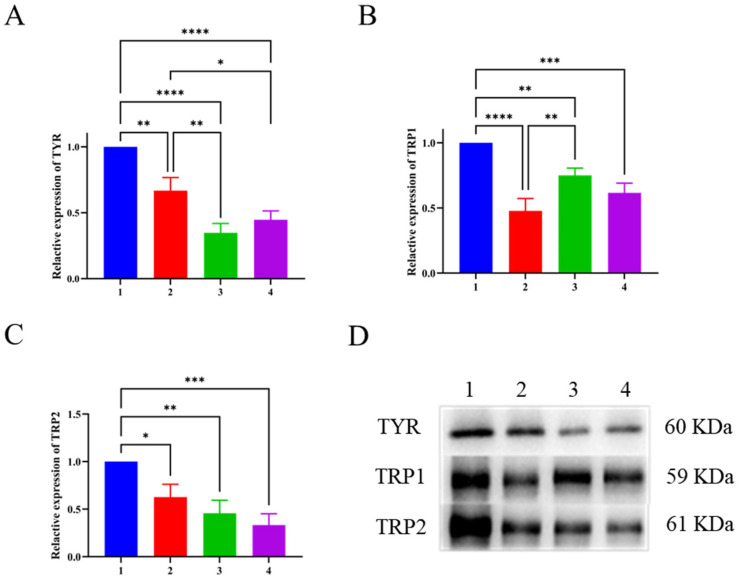
Effects of LFP and kojic acid on TYR (**A**), TRP1 (**B**), and TRP2 (**C**) expressions in mouse melanoma B16 cells and the combined electrophoregrams of three related proteins after different treatments (**D**). The numbers on the abscissas and the electrophoregrams represent groups of the blank control (without LFP, 1), 50 μg/mL LFP treatment (2), 100 μg/mL LFP treatment (3), and 25 μg/mL kojic acid (4). *, **, ***, and **** indicate the significant difference levels at *p* < 0.05, *p* < 0.01, *p* < 0.001, and *p* < 0.0001, respectively.

**Table 1 biomolecules-14-00895-t001:** Identified compounds in LFP with tyrosinase inhibitory activities as reported in the literature.

Code	Compound Name (Literature)	Molecular Formula	Molecular Weight	CAS Number	Retention Time (min)
Flavonoids					
1	Quercetin [18]	C_15_H_10_O_7_	302. 042 0	117-39-5	6.643
2	Isorhamnetin [19]	C_16_H_12_O_7_	316. 057 7	480-19-3	7.108
3	Eriodictyol [20]	C_15_H_12_O_6_	288. 063 0	209-016-4	6.894
Cinnamic acid and its derivatives					
4	Tricoumaroyl spermidine [21]	C_34_H_37_N_3_O_6_	583. 267 4	NA	9.236
5	Dicoumaramide spermidine [21]	C_25_H_31_N_3_O_4_	437. 230 7	NA	9.225
6	Ethyl caffeate [22]	C_11_H_12_O_4_	208. 073 2	102-37-4	9.294
7	2-Hydroxycinnamic acid [23]	C_9_H_8_O_3_	164. 047 2	614-60-8	5.88
8	Coumaric acid [23]	C_9_H_8_O_3_	164. 047 2	614-60-8	10.271
9	trans-4-Methoxycinnamic acid [24]	C_10_H_10_O_3_	178. 062 9	943-89-5	7.181
10	Ferulic acid [25]	C_10_H_10_O_4_	194. 057 7	1 135-24-6	6.894
Benzene and its derivatives					
11	4-Methoxybenzaldehyde [26]	C_8_H_8_O_2_	136. 052 4	123-11-5	5.952
12	p-Anisic acid (4-Methoxybenzoic acid) [26]	C_8_H_8_O_3_	152. 047 2	100-09-4	10.034

## Data Availability

Data are contained within the article and Appendix A.

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
