# Peer review of "Inhibitive Mechanism of Loquat Flower Isolate on Tyrosinase Activity and Melanin Synthesis in Mouse Melanoma B16 Cells"

_biomolecules, 2024, doi:10.3390/biom14080895_

Round 1
Reviewer 1 Report
Comments and Suggestions for Authors
This study provided a preliminary scientific basis for the development of host flower isolates as natural tyrosinase inhibitors and potential whitening or antioxidant ingredients in functional foods and cosmetics. As the authors described, more detailed studies are needed on the melanin synthesis pathways affected by individual main components of LFP and their possible synergistic actions.
The paper is well written, but I would like to make one change.
The following sentence in the Introduction is not accurate, so please change it and cite the paper by Wakamatsu et al. (Pigment Cell Melanoma Research, 2021, 34, 730-747). Tyrosinase does not change tyrosine into dopa, which is then transformed into dopaquinone (DQ), but DQ is directly formed via the oxidation of tyrosine by the action of tyrosinase. DQ undergoes intramolecular cyclization of its amino group to produce cyclodopa (or leucodopachrome), which is then rapidly oxidized by a redox reaction with DQ to give dopachrome (DC) and L-DOPA.
Reviewer 2 Report
Comments and Suggestions for Authors
l Language and Grammar: Review the manuscript for any grammatical errors or awkward phrasings. For example, "Loquat flower isolate product (LFP) was obtained by ethanol extraction and resin purification, and its inhibitory mechanism against TYR activity was investigated by enzyme kinetics and multiple spectroscopy analyses, and the impact on related proteins’ expression in mouse melanoma B16 cells was analyzed using Western blotting." This can be simplified for clarity.
l The abstract contains several long sentences that can be broken down for better readability. For instance, "LFP inhibited TYR activity in a concentration-dependent manner with its IC50 value of 2.8 mg/mL, due to an anti-competitive mechanism by altering the enzyme's conformation." This can be split into two sentences for clarity.
l Ensure that abbreviations such as "LFP" (loquat flower isolate product) are clearly defined at their first occurrence.
l Include more references to previous studies on similar topics to contextualize the research within the existing body of knowledge.
l What are the main components that inhibit tyrosinase, and what are the main components that suppress the expression of tyrosinase?
l In Figure 5, kojic acid is used as a positive control. Generally, kojic acid is known to be an inhibitor of tyrosinase enzyme activity and does not typically affect the expression of tyrosinase. Please discuss this.
Comments on the Quality of English LanguageReview the manuscript for any grammatical errors or awkward phrasings
Reviewer 3 Report
Comments and Suggestions for Authors
The work seems to me appropriate and new, and the analysis has been adequately addressed. More detailed studies are needed on the pathways in melanin synthesis affected by the individual major components in LFP and their possible synergistic effects, but perhaps the authors will perform these at another time.
It is well designed and therefore merits publication to this journal, but not well written with adequate english language and there a few points, which need to be addressed by the authors.
I think you should make revision of the manuscript prior to publication by a native English speaker in order to improve its performance with regard to syntax, grammar and phraseology.
Materyal methods:
Under what conditions were the plants collected and dried? Is there a herbarium record? Who identified the plant?
It should be explained why ethanol and methanol were used in the study and why other solvents were not preferred.
Authors should change the axes of Figure 1 and redefine it.
The resolutıon of the figüres should be improved.
How do you explain the difference between 2 and 3 in trp1 in Western blot?
How would you explain the change between 3 and 4 in Figure 5a,b? Is there any significant difference?
I hope the suggestions for the better of this prepared manuscript will be useful to you.
Comments on the Quality of English Language
The work seems to me appropriate and new, and the analysis has been adequately addressed. More detailed studies are needed on the pathways in melanin synthesis affected by the individual major components in LFP and their possible synergistic effects, but perhaps the authors will perform these at another time.
It is well designed and therefore merits publication to this journal, but not well written with adequate english language and there a few points, which need to be addressed by the authors.
I think you should make revision of the manuscript prior to publication by a native English speaker in order to improve its performance with regard to syntax, grammar and phraseology.
Under what conditions were the plants collected and dried? Is there a herbarium record? Who identified the plant?
It should be explained why ethanol and methanol were used in the study and why other solvents were not preferred.
Authors should change the axes of Figure 1 and redefine it.
The resolutıon of the figüres should be improved.
How do you explain the difference between 2 and 3 in trp1 in Western blot?
How would you explain the change between 3 and 4 in Figure 5a,b? Is there any significant difference?
I hope the suggestions for the better of this prepared manuscript will be useful to you.
Round 2
Reviewer 2 Report
Comments and Suggestions for Authors
Authors are well responded to all comments.
Comments on the Quality of English LanguageIn the title, I suggest "inhibitory mechanism~" instead of "inhibitive ~" .